# Mucosa-Associated Lymphoid Tissue (MALT) Lymphoma in the Gastrointestinal Tract in the Modern Era

**DOI:** 10.3390/cancers14020446

**Published:** 2022-01-17

**Authors:** Eri Ishikawa, Masanao Nakamura, Akira Satou, Kazuyuki Shimada, Shotaro Nakamura

**Affiliations:** 1Department of Gastroenterology and Hepatology, Nagoya University Graduate School of Medicine, Nagoya 466-8560, Japan; makamura@med.nagoya-u.ac.jp; 2Department of Surgical Pathology, Aichi Medical University Hospital, Nagakute 480-1195, Japan; satoakira@aichi-med-u.ac.jp; 3Department of Hematology and Oncology, Nagoya University Graduate School of Medicine, Nagoya 466-8560, Japan; kshimada@med.nagoya-u.ac.jp; 4Department of Gastroenterology, International University of Health and Welfare, Fukuoka 814-0001, Japan; shonaka@iuhw.ac.jp

**Keywords:** extranodal marginal zone lymphoma, *Helicobacter pylori*, gastrointestinal lymphoma, gastric lymphoma, intestinal lymphoma, immunoproliferative small intestinal disease (IPSID), MALT lymphoma

## Abstract

**Simple Summary:**

Extranodal marginal zone lymphoma of mucosa-associated lymphoid tissue (MALT lymphoma), which is frequently linked to chronic antigenic stimulation, is clinically an indolent B-cell lymphoma. The most common site is the stomach, accounting for 35% of MALT lymphomas, with a strong association with *Helicobacter pylori* (*H. pylori*) infection. Although antibiotic eradication is first-line therapy for gastric MALT lymphoma, a recent trend showing increasing *H. pylori*-negative cases may require different treatment strategies. Intestinal MALT lymphoma, including immunoproliferative small intestinal disease, is relatively rare, and the pathogenesis and therapeutic approach have not been fully elucidated. In addition, the gastrointestinal tract was recently considered a preferable site of Epstein–Barr-virus-negative marginal zone lymphoma in the post-transplant setting. We review the updated clinicopathological features, treatment, and evolving concepts of MALT lymphoma of the gastrointestinal tract.

**Abstract:**

Extranodal marginal zone lymphoma of mucosa-associated lymphoid tissue (MALT lymphoma) typically arises from sites such as the stomach, where there is no organized lymphoid tissue. Close associations between *Helicobacter pylori* and gastric MALT lymphoma or *Campylobacter jejuni* and immunoproliferative small intestinal disease (IPSID) have been established. A subset of tumors is associated with chromosomal rearrangement and/or genetic alterations. This disease often presents as localized disease, requiring diverse treatment approaches, from antibiotic therapy to radiotherapy and immunochemotherapy. Eradication therapy for *H. pylori* effectively cures gastric MALT lymphoma in most patients. However, treatment strategies for *H. pylori*-negative gastric MALT lymphoma are still challenging. In addition, the effectiveness of antibiotic therapy has been controversial in intestinal MALT lymphoma, except for IPSID. Endoscopic treatment has been noted to usually achieve complete remission in endoscopically resectable colorectal MALT lymphoma with localized disease. MALT lymphoma has been excluded from post-transplant lymphoproliferative disorders with the exception of Epstein–Barr virus (EBV)-positive marginal zone lymphoma (MZL). We also describe the expanding spectrum of EBV-negative MZL and a close association of the disease with the gastrointestinal tract.

## 1. Introduction

Marginal zone B-cell lymphoma (MZL) is a group of indolent B-cell lymphomas originating from memory B cells normally present in a distinct compartment, known as the marginal zone of secondary lymphoid tissues. MZL is a heterogeneous disease with three different entities recognized in the 2017 World Health Organization (WHO) classification: nodal marginal zone lymphoma (NMZL), splenic marginal zone lymphoma (SMZL), and extranodal MZL of the mucosa-associated lymphoid tissue (MALT lymphoma).

The most common site of MALT lymphoma is the stomach (35%), followed by the ocular adnexa (13%), lungs (8.8%), salivary glands (8.3%), the colorectum (5.2%), and the small intestine (3.4%) [1]. The disease is frequently linked to immune stimulation by bacterial and/or viral agents (chronic hepatitis C virus), autoimmune diseases (Sjögren syndrome or Hashimoto thyroiditis), or IgG4-related disease. The best evidence of an etiopathogenetic link to bacterial agents is provided by the association between *Helicobacter pylori* and gastric MALT lymphoma [2]. Other bacterial infections have been found to be implicated in the pathogenesis of MALT lymphoma arising in the skin (*Borrelia burgdorferi*), the ocular adnexa (*Chlamydophilia psittaci*), lungs (*Achromobacter xylosoxidans*), and the small intestine (*Campylobacter jejuni*). This association is variable in different geographical areas [3,4,5]. Chronic inflammation and/or infection induces the development of oligoclonal and/or monoclonal B-cell populations. In addition, oncogenic events, such as chromosomal alterations and/or translocations, play a relevant role in lymphoma growth and progression to the point that the lymphoproliferative process may eventually become independent of antigenic stimulation.

MALT lymphomas typically have an indolent clinical course [6]. However, approximately 2% of MALT lymphomas undergo histological transformation into aggressive B-cell lymphomas, such as diffuse large B-cell lymphoma (DLBCL), resulting in a worse prognosis [6]. In the gastrointestinal (GI) tract, most gastric MALT lymphomas achieve regression and are effectively cured by the antibiotic eradication of *H. pylori* [7,8]. However, a recent trend showing increasing *H. pylori*-negative gastric MALT lymphoma cases may require different treatment strategies [9,10]. A link between *C. jejuni* infection and immunoproliferative small intestinal disease (IPSID) has also been established, which is classified as a variant of MALT lymphoma, with marked plasma cell differentiation and a favorable response to antibiotic therapy [11,12]. However, in the small and large intestines, only a few reported cases preclude the development of consensus guidelines for MALT lymphoma. Recently, the GI tract has been focused on as the preferable site of MALT lymphoma in the post-transplant setting [13].

In this review, we focus on the recent advances in the diagnosis and treatment of MALT lymphoma affecting the GI tract.

## 2. Clinicopathological Features of GI Lymphomas

In our experience of 455 patients with GI lymphomas in Kyushu University Hospital, the most common involved site was stomach (342 cases; 75%), followed by the small and large intestines (96 cases; 21%) [14]. The remaining 17 cases had lymphomas both in the stomach and intestines (4%).

### 2.1. Gastric Lymphomas

The most common histologic type of gastric lymphoma was MALT lymphoma (170 cases, 50%), followed by DLBCL (129 cases, 38%) [14]. Primary gastric MALT lymphoma frequently occurs in patients aged 50–60 years [7,8]. Cases newly diagnosed as gastric MALT lymphoma have decreased in the last decade, which is likely associated with a decline in the prevalence of *H. pylori* infection among healthy population and/or better control of *H. pylori* among infected people [15,16]. The association between *H. pylori* and gastric MALT lymphoma has decreased recently, particularly in the Western world, with instances of *H. pylori*-positive cases reported to be as low as 33% [17]. Patients with gastric MALT lymphoma are usually asymptomatic and found by screening esophagogastroduodenoscopy (EGD), with approximately 90% of cases having clinical stage I disease [7,8]. Bone marrow involvement is exceedingly rare (3%) [18]. There is an increasing trend of *H. pylori*-negative gastric MALT lymphoma cases, accounting for 6–40% of recent cases [7,8,9,19,20,21,22,23,24,25,26,27,28]. Compared to *H. pylori*-positive cases, *H. pylori*-negative gastric MALT lymphomas tend to be in an advanced clinical stage [24]. Approximately 60% of gastric MALT lymphomas can be detected by fluorine-18-fluorodeoxyglucose (^18^F-FDG) positron emission tomography (PET) or PET/computed tomography (CT) [29].

DLBCL is the second-most-common histologic type in gastric lymphomas. *H. pylori* infection was observed with and without the MALT lymphoma component (50–70% and 25–40%, respectively).

### 2.2. Intestinal Lymphomas

MALT lymphoma of the small intestine is relatively rare, with various symptoms, including abdominal pain, bloating, constipation, and diarrhea [30,31,32]. IPSID is classified as a variant of MALT lymphoma, with marked plasma-cell differentiation, and is also known as an α heavy chain disease (αHCD) [12,33]. IPSID has been reported to affect young adults (mean age 32 years) of a low socio-economic status living in poor hygienic conditions. IPSID was also recently diagnosed in renal and liver transplant recipients [34,35]. Malabsorption, intermittent diarrhea, and abdominal pain are the most frequent symptoms [33]. Serologically, roughly half of IPSID patients have elevated IgA levels, and the presence of anomalous α heavy chain protein is detected in 20% to 90% of patients [33,36].

The pathogenesis of colorectal MALT lymphoma is not well understood. Approximately 20% of patients with colorectal MALT lymphoma are *H. pylori* positive, and other chronic inflammation has not been documented except for a case of rectal MALT lymphoma that was possibly associated with primary biliary cirrhosis and secondary Sjögren syndrome [37,38,39]. MALT lymphoma of the large intestine affects individuals in their fifth through seventh decades, with a female predominance [38,40,41]. The ileocecum and the rectum are commonly affected [40,42]. Of the cases, 75% involved the rectum, including simultaneous involvement of the cecum in one-fifth of these patients [37,38]. Patients with colorectal MALT lymphoma sometimes present with abdominal discomfort, diarrhea, constipation, or a positive fecal occult blood test [40,43]. Approximately 90% of colorectal MALT lymphomas are localized clinical stage Lugano I-II, and bone marrow involvement is rarely seen (4–9%) [37,40,41].

## 3. Pathogenesis in Relation to Bacterial Infection or Chronic Antigenic Stimulation

*H. pylori* infection could result in the accumulation of MALT, from which gastric MALT lymphoma arises, though MALT as typified in Peyer’s patches is absent in the normal stomach. Proliferation of gastric MALT lymphoma in response to *H. pylori* is due to the recognition of *H. pylori* by tumor-infiltrating T cells [44]. This involves cognate interaction between B and T cells, and bystander T cells help the interaction via soluble ligands and cytokines, such as CD40 ligand and B-cell-activating factor (BAFF), activating the non-canonical nuclear factor (NF)-κB pathway [45].

In *H. pylori*-negative cases, non-*H. pylori* Helicobacters (NHPHs; also referred to as *H. heilmannii*) are occasionally associated with gastric MALT lymphoma [46]. Takigawa et al. reported that 55% of patients with *H. pylori*-negative gastric MALT lymphoma without a *BIRC3-MALT1* chimeric transcript are infected with an NHPH [20]. NHPHs typically have a large spiral-shaped morphology and comprise *H. suis, H. felis, H. bizzozeronii, H. salomonis,* and *Candidatus H. heilmannii*, accounting for only 0.1% of cases in which gastric biopsy is performed [47,48]. A higher rate (21%) of NHPH infection was shown by a PCR of the *H. pylori*-negative cases with gastric disease, mainly including gastric MALT lymphoma, chronic or nodular gastritis, and gastroduodenal ulcer [49]. *H. suis* is the most prevalent NHPH species in human. Yang et al. indicated that IFN-γ produced by B cells plays an important role in the formation of gastric lymphoid follicles after *H. suis* infection [50].

Chronic antigenic stimulation with *C. jejuni* is known to lead to gut-mucosa-associated, IgA-producing plasma cell proliferation. IPSID shares the feature of mutant immunoglobulin heavy chain production, characterized by the deletion of the N-terminus encompassing the variable heavy chain (V_H_) and the constant heavy chain (CH1) domains [33]. In the absence of the CH1 domain, the mutant heavy chain cannot assemble with a light chain to produce an intact immunoglobulin. Therefore, the disease secretes a monotypic truncated immunoglobulin α heavy chain lacking an associated light chain.

## 4. Molecular Pathology

Chromosomal translocations associated with MALT lymphomas include t(11;18)(q21;q21), resulting in the production of a chimeric protein (BIRC3-MALT1), and t(1;14)(p22;q32), t(14;18)(q32;q21), and t(3;14)(p14.1;q32), which result in transcriptional deregulation of BCL10, MALT1, and FOXP1, respectively [51]. These different translocations are linked by the roles of *BCL10* and *MALT1* in the NF-κB molecular pathway in lymphocytes [45]. t(11;18)(q21;q21)/*BIRC3(API2)-MALT1* translocation occurs mostly in gastric MALT lymphomas, accounting for 7% to 21% of cases, which are often *H. pylori* negative but positive for nuclear BCL10. However, t(11;18)/*BIRC3(API2)-MALT1* translocation has been detected in 10–42% of intestinal MALT lymphoma cases [52,53].

In *H. pylori*-positive gastric MALT lymphomas unresponsive to eradication therapy, NF-κB pathway mutations, including TNF-receptor-associated factor 3 (*TRAF3*) mutations and TNF-alpha-induced protein 3 (*TNFAIP3*) mutations, and *MALT1* translocations are enriched [54]. Furthermore, frequent *TNFAIP3* inactivating mutations and translocations in *MALT1/IGH* have been detected in *H. pylori*-negative gastric MALT lymphoma, suggesting that the NF-κB pathway can be a major driver in the pathogenesis of this group [10]. In addition, a recent transcriptome analysis showed that long noncoding RNAs *GHRLOS* and another 44 mRNAs are aberrantly expressed in gastric MALT lymphoma [55]. Other translocations, t(1;14)(p22;q32)/*IGH-BCL*10 and t(3;14)(p14;q32)/*IGH-FOXP1*, occur infrequently in gastric and intestinal MALT lymphomas [45,56].

## 5. Pathological Findings

MALT lymphoma is defined as an extranodal lymphoma composed of morphologically heterogeneous small B cells, including marginal zone (centrocyte-like) cells; monocytoid cells; small lymphocytes; and scattered large cells, such as immunoblasts and centroblast-like cells. Centrocyte-like cells are small- to intermediate-sized cells with a slightly irregular nucleus [56,57]. Clear cells are regarded as a variant of centrocyte-like cells, with an abundant pale cytoplasm. A proportion of cases are accompanied by plasma cell differentiation. Lymphoma cells occasionally invade the epithelium, forming lymphoepithelial lesions (LELs), which are aggregates of three or more marginal zone cells with destruction of the glandular epithelium. The presence of LELs is suggestive of MALT lymphoma in the stomach, but this is not essential for this diagnosis because they can be also seen in other low-grade B-cell lymphomas. In immunohistochemistry, MALT lymphoma cells are CD20+, CD79a+, BCL2+, BCL6−, CD5−, CD10−, and CD23−. A distinction from follicular lymphoma (positive for CD10 and BCL6), mantle cell lymphoma (CD5, cyclin D1, and SOX11), and chronic lymphocytic leukemia (CD5, CD23, and LEF1) is needed.

In cases with IPSID, a dense lymphoplasmacytic infiltrate confined to the mucosa and/or the submucosa is seen in early stages, which may lead to villous blunting or atrophy of the small intestine [58]. The small lymphomatous cells are positive for CD19 and CD20 but negative for CD5, CD10, and CD23. The plasmacytic cell component may only express plasma cell markers, such as CD138 [56]. Both lymphocytes and plasma cells express IgA in the cytoplasm, but not kappa and lambda, immunoglobulin light chain markers [12].

Distinguishing MALT lymphoma from localized lymphoid hyperplasia (LLH) is occasionally challenging. In contrast to MALT lymphoma, LLH is usually characterized by large lymphoid follicles with active germinal centers and a narrow mantle zone and marginal zone. Most LLHs have a benign course and are resolved [59]. The distinction between MALT lymphoma and follicular lymphoma can be problematic, particularly if the biopsy is small and the MALT lymphoma shows follicular colonization associated with up-regulation of CD10 and/or BCL6 in the intrafollicular compartment. In these instances, fluorescent in situ hybridization (FISH) for the t(14;18)/*BCL2* gene rearrangement may be helpful.

In gastric MALT lymphoma, diagnosis of a small biopsy specimen using Wotherspoon histological scoring is relevant to show the degree of confidence in making a diagnosis [2]. Because of the difficulty in applying the Wotherspoon system and its low reproducibility, pathologists of Groupe d’Etude des Lymphomes de l’Adulte (GELA) established a post-treatment histological grading system that classifies the morphological features into four categories: complete histological response (ChR), probable minimal residual disease (pMRD), responding/residual disease (rRD), and no change (NC) [60]. Patients with ChR or pMRD are regarded as having a clinical CR and responders, whereas the remaining categories (rRD or NC) are considered to be non-responders.

## 6. Endoscopic Findings

The endoscopic findings in MALT lymphoma are mainly classified as superficial type or other, which includes mass-forming, protruding, and diffuse infiltrating types [7,61]. A superficial appearance is the most common endoscopic type, accounting for approximately 70–80% of cases [7,8,24]. Therefore, gastric MALT lymphoma is often indistinguishable from gastric cancer or gastritis based on endoscopic findings alone. Nakamura et al. originally revealed that *H. pylori*-negative cases are more frequently located in the proximal stomach and invade the submucosa or beyond but less frequently appear as the common superficial type compared to *H. pylori*-positive lymphomas [62]. In addition, *H. suis*-associated gastric MALT lymphoma was recently documented to have a nodular gastritis-like appearance, which is distinct from the common type [63]. The endoscopic findings on narrow band imaging (NBI) magnifying endoscopy were also characterized by a tree-like appearance (TLA), which corresponded to abnormally large vessels resembling a tree trunk with long, bare branches [64]. Moreover, endocytoscopy may be useful for detecting intra-glandular aggregation of cellular components and small to moderately sized nuclei in gastric MALT lymphoma, but further studies are needed [65].

In small intestinal MALT lymphoma, single- or multiple-ulcerated lesions or polypoid lesions without villi are often identified by endoscopic examination (Figure 1) [66,67]. Some cases present with changes in the form of lesions within a short time period. Therefore, re-examination may lead to reliable diagnosis [30]. However, small intestinal MALT lymphoma, sometimes, may not form a visible tumor mass in routine practice because of complicated small intestinal obstruction [31,32]. Interestingly, a color-faded cobble-stone-like erosion in the ileum was reported in a rare case of MALT lymphoma showing histologically extensive plasma cell differentiation with prominent Russell bodies [68].

Polypoid or nodular lesions are identified in two-thirds of the patients with IPSID; the other one-third of the patients present with erythematous mucosal thickening or mucosal edema (Figure 2). The infiltrative and nodular patterns are the most sensitive and specific for a diagnosis of αHCD [58]. Kurimoto et al. reported capsule endoscopic findings of a slightly rough mucosa with swelling of the villi within the small intestine and a scar from a longitudinal ulcer extending from the superior jejunum to the middle ileum in patients with IPSID [69].

Colorectal MALT lymphoma can have varying endoscopic appearance, such as flat, elevated, polypoid, or semipedunculated lesions, the surface of which is typically smooth, granular, or nodular (Figure 3) [37,43,70,71]. In the rectum, a polypoid lesion has been reported to be 10-fold more common than an ulcerative lesion [37]. The tumors also vary in size, with the majority measuring between 15 and 30 mm [39,41,43,72]. A magnified endoscopy of the tumor surface shows a type I pit pattern with dilated tree-like microvessels on NBI [73]. Endocytoscopy could detect diffuse interglandular infiltration of small atypical cells [74].

## 7. Treatment and Clinical Response

### 7.1. Antibiotic Therapy

Eradication therapy for *H. pylori* is now used as the first-line therapy for gastric MALT lymphoma, regardless of stage. Antibiotic eradication of *H. pylori* leads to regression and an effective cure of gastric MALT lymphoma in approximately 70–80% of patients [7,20,26,27]. Triple-therapy regimens combining a proton-pump inhibitor for 4 weeks plus clarithromycin with either amoxicillin or metronidazole for 10–14 days are highly effective [75]. Most patients with minimal histologically residual gastric MALT lymphoma following *H. pylori* eradication exhibit CR after more than 12 months [76], some of whom achieve CR after more than 24 months [7,22,57,77]. Therefore, it is recommended to wait for at least 12 months before starting another treatment to avoid overtreatment of patients without progression [57].

Although gastric MALT lymphoma has a favorable prognosis, the remission status seems to affect progression-free survival (PFS). Kiesewetter et al. reported that responders (i.e., ChR/pMRD/rRD) to first-line treatment had a longer PFS than patients with stable disease (SD) in gastric MALT lymphoma [19].

The predictors of resistance to *H. pylori* eradication include the absence of *H. pylori*, advanced stage, proximal location, endoscopic non-superficial type, polypoid appearance, deep submucosal invasion by endoscopic ultrasonography, and the presence of t(11;18)/*BIRC3*(*API2)-MALT1* translocation [20,75,78,79,80,81].

Although a significant decrease in the rate of gastric MALT lymphoma associated with *H. pylori* infection has been reported [17], the effectiveness of eradication therapy can be seen even in *H. pylori*-negative cases, accounting for 0–57%, though the CR rate is lower than that of *H. pylori*-positive patients [8,28,62,82]. A meta-analysis by Jung et al. showed that the CR rate after eradication therapy was 29% in patients with *H. pylori*-negative gastric MALT lymphoma [83]. Studies of patients with *H. pylori*-negative gastric MALT lymphoma are summarized in Table 1 [7,8,9,10,20,21,22,23,24,25,27,28]. In patients with *H. pylori*-negative gastric MALT lymphoma who receive eradication therapy, the median time to CR is longer than in *H. pylori*-positive patients [8]. Moreover, 75% of NHPH-positive cases have been reported to achieve CR [20]. In small intestinal MALT lymphoma, the ineffectiveness of *H. pylori* eradication has been documented in a handful of studies [30,67,84]. Although there are few cases of duodenal MALT lymphoma reported in the English literature, Na et al. reported that 46% of patients with duodenal MALT lymphoma were *H. pylori* positive and 50% of cases treated with eradication therapy achieved CR [85]. Antibiotic therapy for at least 6 months, including ampicillin, metronidazole, and tetracycline, alone or in combination, should be attempted in patients with IPSID [12]. Tetracycline yielded CR rates of 30% to 70%, lasting months to several years, in patients with early disease [33].

In colorectal localization, some studies have shown regressed cases of MALT lymphoma following *H. pylori* eradication [86,87]. A review of the English literature focused on rectal MALT lymphoma reported that 37% of cases showed *H. pylori* eradication with various regimens, regardless of infection status, and 37% required second-line therapy [41]. Antibiotics may eradicate other unknown organisms involved in the development of colorectal MALT lymphoma.

### 7.2. Watch-and-Wait Strategy and Re-Eradication after Eradication Therapy

In gastric MALT lymphoma, non-responders without progressive disease and recurrent patients after eradication therapy sometimes require a watch-and-wait approach without any additional treatment (Table 2). Nakamura et al. reported that the watch-and-wait approach resulted in no change in 20% of non-responders with gastric MALT lymphoma after eradication therapy [7]. In addition, more than half of recurrent cases with gastric MALT lymphoma under the watch-and-wait regimen achieved CR [7,8,24]. Recurrence of gastric MALT lymphoma was occasionally associated with reinfection with *H. pylori*, and more than 80% of the cases achieved CR after re-eradication treatment [8,22,24].

An interesting case with MALT lymphoma in the terminal ileum has also been reported. The case presented with multiple protruding lesions and experienced spontaneous regression in preoperative endoscopy [88]. In colorectal MALT lymphomas, Jeon et al. reported that four patients in stage I and one patient in stage IV, without any specific treatment because of their conditions and underlying diseases, experienced no disease progression during the follow-up period, which may be based on the favorable prognosis of low-grade lymphoma [40].

### 7.3. Radiotherapy

Non-responders to eradication therapy have been detected in 17–61% of patients with gastric MALT lymphoma (Table 2) [7,8,9,24,26,27,28]. Several studies have acknowledged the effectiveness of radiotherapy (RT) or immune-/chemotherapy as second-line treatment for non-responders with gastric MALT lymphoma [8,24,26,28]. Moderate-dose RT appears to be effective in patients with localized gastric MALT lymphoma persisting after *H. pylori* eradication [89]. In addition, 25 patients with localized disease receiving RT had a favorable outcome, with a 10-year recurrence-free rate of 92% [90]. The largest cohort, reported by Yahalom et al., comprised 178 *H. pylori*-independent gastric MALT lymphoma patients, 95% of whom exhibited a complete pathological response to RT (median dose 30 Gy), with rare acute or late toxicity [91]. Schmelz et al. revealed that a reduced RT dose of 25.2 Gy was as effective as the standard RT of 36 Gy in localized gastric MALT lymphoma [92]. The German Study Group on Gastrointestinal Lymphoma (DSGL) reported that stage-adopted reduction from extended field RT (EFRT) to involved field RT (IFRT) led to a better outcome and reduction of adverse effects in gastric MALT lymphoma [93].

Although RT is effective for MALT lymphoma limited to a focal area of the GI tract, including the rectum [43,94], precise targeting of the small intestine by RT is difficult because it is not fixed in the abdominal cavity. Reinartz et al. showed that RT adapted to stage, histology, and resection in multimodal treatment of intestinal lymphoma achieves excellent local tumor control and survival rates despite the partially decreasing field size [95]. In addition, current involved site RT (ISRT) offers the option of further reduction of normal tissue complication probability and a clinical trial of ISRT 20Gy for indolent localized gastrointestinal lymphoma is ongoing.

Recently, very-low-dose RT (4Gy) was also shown to be effective in MZLs, including those at extranodal sites, which may alternatively reduce toxicities and the duration of treatment [96].

### 7.4. Chemo/Immunotherapy

Rituximab as a single agent is known to exert significant antitumor activity in MALT lymphoma. Combination therapy with rituximab and alkylating agents in patients with gastric MALT lymphoma has greater efficacy than monotherapy with each, but some patients suffer from considerable toxicity [97]. Rituximab in combination with chlorambucil has also been shown to achieve CR rates of 91% and improve event-free survival (EFS) with little added toxicity in gastric MALT lymphoma patients with disease progression any time post *H. pylori* eradication or in gastric MALT lymphoma patients with SD and persistent lymphoma more than 1 year after *H. pylori* eradication (5-year EFS, 77%) [98]. Furthermore, gastric MALT lymphoma patients with unequivocally active disease after failure of *H. pylori* eradication or in subsequent relapse have been reported to have excellent CR rates, of 100%, and 7-year EFS of 89.5% when treated with the combination of rituximab and bendamustine, which is superior to treatment with the combination of rituximab and chlorambucil [99]. Some patients with *H. pylori*-negative gastric MALT lymphoma receive RT or chemotherapy as the first-line treatment, resulting in a high CR rate, of more than 80% (Table 1) [24,25,28].

For small intestinal MALT lymphoma with multiple lesions, no response to eradication, or extending throughout the entire small intestine, chemo-immunotherapy has been reported to be effective (Table 3) [30,66,67,84,100,101]. A large case analysis of colorectal MALT lymphoma reported that chemotherapy is most commonly performed for advanced disease [40]. Furthermore, IPSID patients without marked improvement after a 6-month course of antibiotics, without CR within 12 months, or with an intermediate or advanced stage of disease should also receive anthracycline-based combination chemotherapy, as this has been shown to result in a CR rate of approximately 64% in the latter group [102,103].

### 7.5. Endoscopic Resection

A summary of recently reported cases of colorectal MALT lymphoma is shown in Table 4. In patients with colorectal MALT lymphoma, endoscopic resection in cases with localized disease has been noted to usually achieve CR [43,70,73,106,107,108]. Approximately 40% of patients with colorectal MALT lymphoma receive endoscopic resection, 20% of whom are treated with additional RT because of positive resection margins [40]. Although one patient with transverse colon disease has been reported to have experienced recurrence at the terminal ileum, most patients treated with endoscopic mucosal resection (EMR) experience no progression. In addition, for residual tumors after EMR, endoscopic submucosal dissection (ESD) may be effective as salvage therapy [109]. In patients with colonic MALT lymphoma treated with RT, an obstructing mass could shrink and subsequently be resected by an endoscopic procedure [110].

### 7.6. Surgery

Surgical treatment is now restricted to gastric MALT lymphoma cases with rare complications, such as bleeding or perforation, that cannot be controlled endoscopically [57]. Patients with small intestinal MALT lymphoma causing a dilated segment, obstruction, or perforation undergo surgical resection, some of whom receive additional chemo-/immunotherapy (Table 3) [31,32,104,105]. Colonic MALT lymphoma cases of locally extended masses occasionally undergo surgical resection [43,71].

### 7.7. Outcome

Recurrence occurs in 3–21% of patients with gastric MALT lymphoma after eradication therapy [7,8,22,24,27,28], which may even occur 7 years after CR [22]. Patients with gastric MALT lymphoma have favorable outcomes, with 10-year disease-free survival and OS of 79–86% and 83–95%, respectively [7,22,27]. Although small intestinal MALT lymphoma has a good prognosis among patients with primary small intestinal lymphoma [111], only a small number of cases with small intestinal MALT lymphoma have been analyzed and the prognosis of the tumor remains to be elucidated. Patients with colorectal MALT lymphoma have favorable outcomes, with 5-year PFS of 92% and 5-year OS of 94% [40]. In the literature review of 73 patients with colorectal MALT lymphoma, the overall remission rate and the tumor recurrence rate were 96% and 7%, respectively [38].

## 8. The Risk of Developing Carcinoma and Other Malignancies

Capelle et al. reported a higher risk of synchronous and metachronous gastric cancer in patients with gastric MALT lymphoma [15]. In their study, gastric cancer was diagnosed in 2.4% of 1419 patients with gastric MALT lymphoma, the risk of which was 6-fold in comparison to the general population. In addition, many or most patients with gastric MALT lymphoma who developed gastric cancer were diagnosed with adenocarcinoma concomitantly (53%) or during later surveillance (38%). Almost all reported cases with adenocarcinoma were in early stages [26,112], though Isaka et al. reported rapid development of advanced adenocarcinoma with *BIRC3(API2)*-*MALT1* chimeric transcript-positive gastroduodenal MALT lymphoma as a collision tumor [113]. We also reported a collision tumor of advanced gastric adenocarcinoma and MALT lymphoma detected 16 years after diagnosis of splenic MZL [114].

The development of metachronous neoplasms other than gastric adenocarcinomas, including various solid tumors and hematological neoplasms, has also been reported [7,27]. In addition, synchronous cases of adenocarcinoma and MALT lymphoma in the colorectum have been reported, but the correlation between the two diseases remains unclear [115].

## 9. Epstein–Barr Virus (EBV)-Negative Post-Transplant EMZL in the GI Tract

Indolent B-cell lymphoma has traditionally been excluded from the category of monomorphic post-transplant lymphoproliferative disorders (PTLDs). PTLDs are a heterogeneous group of lymphoid or plasmacytic proliferations that develop secondary to chronic immunosuppression after solid-organ or allogeneic hematopoietic stem-cell transplantation. Most PTLDs are derived from B cells and are associated with EBV infection [116]. Post-transplant MALT lymphoma is generally gastric, with a strong correlation with *H. pylori* infection, and responds well to eradication therapy alone, which is similar to what is seen in immunocompetent hosts [117,118,119]. Therefore, only EBV-positive MZL has been considered a form of PTLD. However, the spectrum of EBV-negative MZL in the post-transplant setting has recently been expanded.

Galera et al. reported cases with post-transplant EBV-negative MZL, all arising in solid-organ transplant recipients [13]. Five of seven patients with EMZL presented with intestinal disease, which included the colon in four cases, suggesting a difference in the preferential involvement of the intestine between post-transplant EMZL and sporadic cases. All of the patients had past histories of treatment with immunosuppressive agents and were diagnosed more than 1 year after transplant, and two had a history of ulcerative colitis. The endoscopic findings varied, including multiple polyps, erythema, ulcers, and masses. Most patients were treated with reduced immunosuppression and rituximab. Although chronic antigenic stimulation has been postulated to play a role in the pathogenesis of EMZL, the transplanted organ may fulfill this role. This issue should be investigated more in the future.

## 10. Conclusions

Many patients with gastric MALT lymphoma respond to *H. pylori* eradication, although the association between *H. pylori* and gastric MALT lymphoma has decreased markedly recently. Antibiotic therapy has also been shown to be effective for a subset of patients with *H. pylori*-negative gastric MALT lymphoma, including NHPH-associated cases. Cases with chromosomal rearrangement and/or genetic alterations should be distinguished because of a high frequency of non-responders. Gastric MALT lymphoma patients generally experience a favorable clinical course, even non-responders and recurrent cases treated with RT or immunochemotherapy. However, surveillance endoscopy is recommended after CR considering the possibility of metachronous gastric cancer. IPSID, also known as a variant of small intestinal MALT lymphoma involving *C. jejuni*, is uncommon, and early-stage patients may derive a benefit from antibiotic treatment. Colorectal MALT lymphoma is rare, with varying endoscopic appearance, including elevated polypoid lesions. The pathogenesis is not well known, and treatment approaches vary. The efficacy of antibiotic therapy is still controversial, but endoscopic procedures tend to be considered in endoscopically resectable colorectal MALT lymphoma with localized disease. Recently, preferential involvement of the intestine in EBV-negative MZL arising in the setting of immunosuppression was noted. Further studies of MALT lymphoma focusing on the intestines will be of interest and may lead to the development of a treatment strategy.

## Figures and Tables

**Figure 1 cancers-14-00446-f001:**
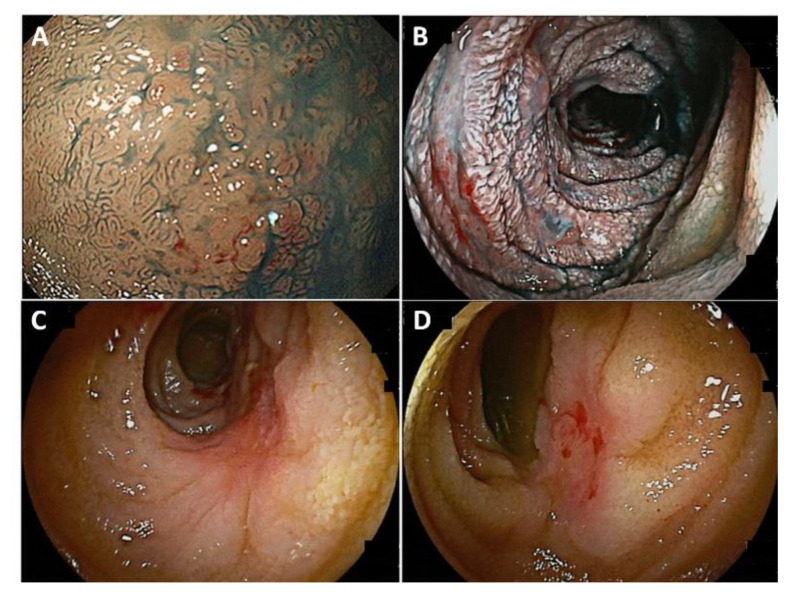
Endoscopic images of small intestinal MALT lymphoma. (**A**) Irregularity of the duodenal mucosa. (**B**) A shallow ulcer with diffuse mucosal edema in the jejunum. (**C**,**D**) Multiple ulcerated lesions in the ileum.

**Figure 2 cancers-14-00446-f002:**
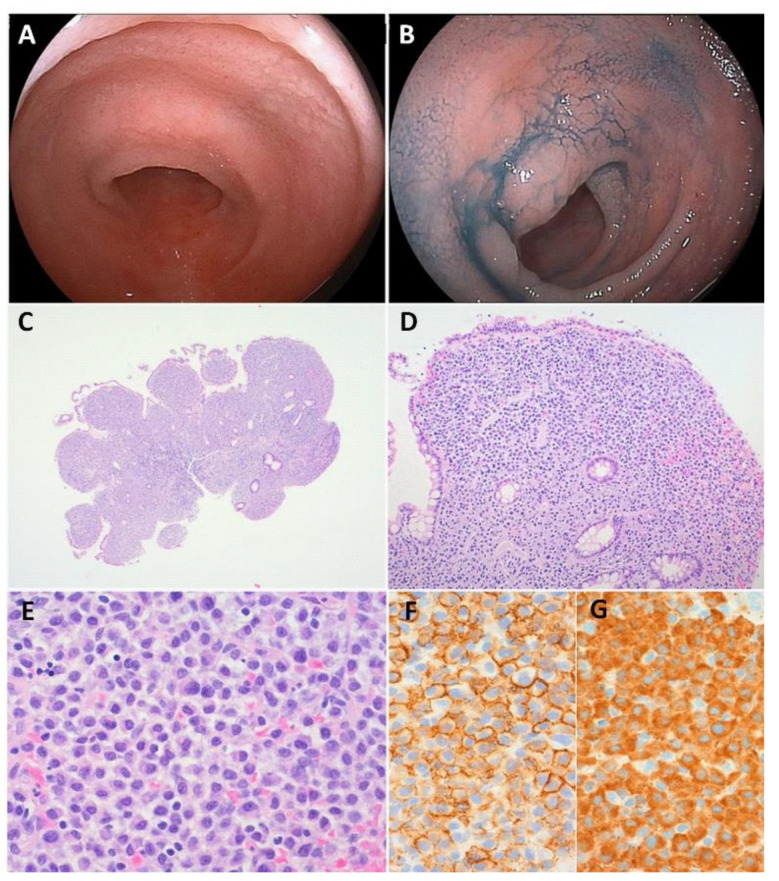
Immunoproliferative small intestinal disease (IPSID). (**A**,**B**) Double-balloon endoscopy showing villous edema in the ileum. (**C**) A biopsy specimen revealing blunting of the villi. (**D**,**E**) A dense lymphoplasmacytic infiltrate detected. (**F**,**G**) The tumor cells are positive for CD138 (**F**) and IgA (**G**). Original magnification: ×25 (**C**), ×100 (**D**), and ×400 (**E**–**G**).

**Figure 3 cancers-14-00446-f003:**
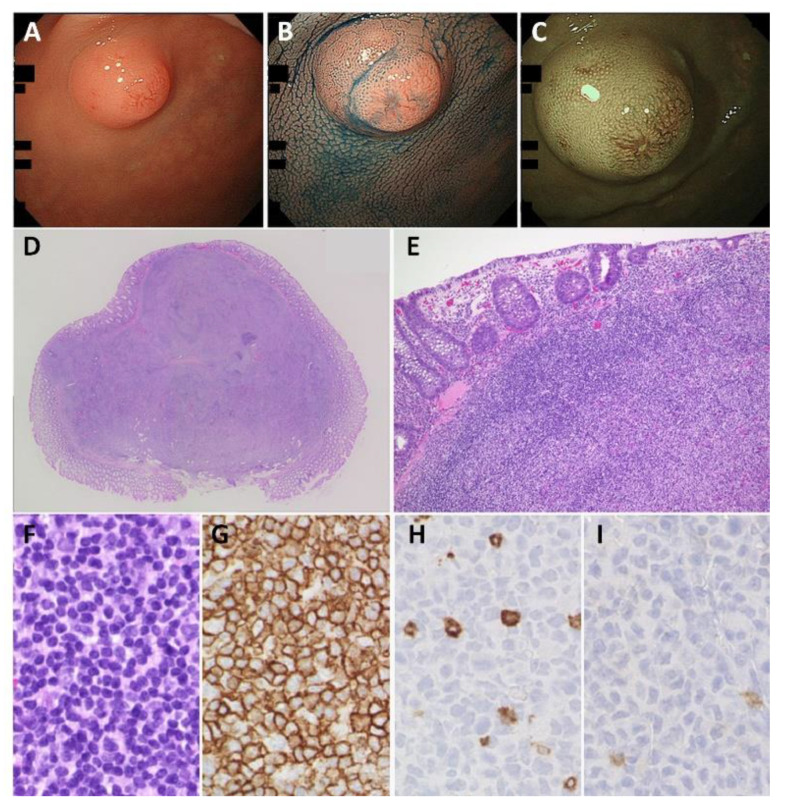
Rectal-mucosa-associated lymphoid tissue (MALT) lymphoma. (**A**,**B**) Endoscopic images showing a 10 mm solitary elevated lesion with erosion on the surface. (**C**) Narrow band imaging showing enlarged vessels on the surface of the lesion. (**D**) Specimen from endoscopic mucosal resection. (**E**,**F**) Monotonous proliferation of small- to intermediate-sized centrocyte-like cells observed. (**G**–**I**) The tumor cells are positive for CD20 (**G**) and negative for CD5 (**H**) and CD10 (**I**). Original magnification: ×100 (**E**) and ×400 (**F**–**I**).

**Table 1 cancers-14-00446-t001:** *H. pylori*-negative patients with gastric MALT lymphoma.

*n*	Stage	*BIRC3-MALT1*	1st-Tx				2nd-Tx after Eradiation		Ref.
			Eradication		Other Tx				
			*n*	CR Rate	*n*	CR Rate	*n*	CR Rate *	
14	I, II1	NE	14	33%					[27]
18	I, II1	7 (64%)	17	29%	1 (RT)	100%	12 (WW 7, RT 5)	100%	[25]
44	I–IV	NE	44	14%					[7]
13	I, II1	5 (56%)	5	40%	5 (RT)	80%	3 (WW 1, RT 1, CTx 1)	100%	[24]
24	I, II	3 (23%)	13	38%			8 (CTx ± IM 8)	100%	[23]
28	I–IV	NE	28	57%			12 (WW 5, RT 6, CTx 1)	71%	[8]
30	I, II	NE	18	33%					[22]
25	I, II1	7 (28%)	25	36%			14 (RT 3, CTx ± IM 11)	86%	[21]
131	I–IV	NE	63	17%	68 (RT or CTx ± IM or S)	72%			[9]
57	I–IV	22 (39%) ^†^	9	0%	48 (WW 2, RT 1, CTx 44, other 1)	33%			[10]
34	NE	NE	34	44%			19 (RT 19)	100%	[20]
37	I–IV	11 (30%)	18	11%	19 (RT 16, CTx 3)	89% ^‡^	16 (WW 3, RT 12, CTx 1)	92%	[28]

CR: complete remission; CTx: chemotherapy; IM: immunotherapy; NE: no evaluation; RT: radiotherapy; S: surgery; Tx: treatment; WW: watch and wait. * CR rate in the treated patients except for the cases under the watch-and-wait strategy; ^†^ MALT1 rearrangement; ^‡^ 100% in RT, 33% in CTx.

**Table 2 cancers-14-00446-t002:** Second-line treatment in non-responders to eradication therapy in gastric MALT lymphoma.

*n*	Stage	*HP* (−)	Non-CR, *n* (%)		2nd-Tx						2nd-Tx *		Ref.
					WW		RT		Other Tx				
					*n*	CR Rate	*n*	CR Rate	*n*	CR Rate	*n*	CR Rate	
105	I, II1	13%	24	(24%)	10	NE			14 (CTx ± IM 12, CTx + S 1, S 1)	27%			[27]
60	I, II1	12%	10	(17%)	1	0% (SD 1)	7	100%	2 (RT + CTx 1, ER 1)	100%			[26]
420	I–IV	10%	97	(23%)	15	0% (NC 15)					82 ^†^	94%	[7]
66	I, II1	20%	16	(29%)	2	50%	9	89%	1 (CTx)	0%			[24]
345	I–IV	8%	61	(18%)	42	NE	17	88%	1 (CTx)	100%			[8]
339	I–IV	40%	157	(61%)	57	NE					100 ^‡^	79%	[9]
96	I–IV	39%	44	(60%)	3	NE	33	91%	5 (CTx)	63%			[28]

CR: complete remission; CTx: chemotherapy; ER: endoscopic resection; HP: Helicobacter pylori; IM: immunotherapy; NC: no change; NE: no evaluation; R: rituximab; RT: radiotherapy; S: surgery; SD: stable disease; Tx: treatment; WW: watch and wait. * CR rate of each treatment was unavailable. ^†^ RT 47, CTx 22, RT + CTx 5, S 5, ER 1, R 1, and third-line antibiotic therapy 1; ^‡^ RT, CTx ± IM, and Sur were performed but detailed information was unavailable.

**Table 3 cancers-14-00446-t003:** Summary of recently reported cases of small intestinal MALT lymphoma.

Age	Sex	Site	Macroscopic Findings	Stage	1st-Tx	Response	FU Time, years	Ref.
75	M	Jejunum	Stricture, shallow ulcer	NA	E, R-CTx	PR	NA	[30]
78	F	Jejunum	Multiple ulcerative lesions	II1	S (perforation)	CR	NA	[104]
67	F	Ileum	Long raised mucosal surface	II2	S (obstruction)	CR	0.5	[32]
58	M	Ileum	8 cm saccular dilation	IIIE	S + R	PR	0.5	[31]
55	M	Ileum	NA	NA	S + CTx (dilated segment)	CR	1	[105]
56	F	Ileum	NA	NA	R-CTx	CR	NA	[101]
73	F	Ileum	Cobble-stone-like erosion	NA	None (progression to T-prolymphocytic leukemia)	NA	0.6 (DD)	[68]
35	F	Ileum	Multiple tumors and ulcers	Lo	CTx	CR	NA	[66]
38	M	TI	Multiple protruding lesions	Lo	None *	CR	2	[88]
61	F	TI	Multiple polypoid lesions	II	A, CTx	NA	NA	[67]
73	F	Entire	Nodular mucosal lesions	I	R-CTx	CR	NA	[100]
50	F	Entire	Multiple polypoid lesions	II	A, R-CTx	CR	5	[84]

A: antibiotics; CR: complete remission; CTx: chemotherapy; DD: died of disease; E: eradication; F: female; FU: follow-up; I: ileum; J: jejunum; Lo: localized; M: male; NA: not available; PR: partial response; R: rituximab; S: surgery; TI: terminal ileum; Tx: treatment. * Spontaneous regression in 2 months.

**Table 4 cancers-14-00446-t004:** Summary of recently reported cases of colorectal MALT lymphoma.

Age	Sex	Site	Number	Size, mm	Stage	1st-Tx	Response	FU Time, years	Ref.
*n* = 51		Re, C, IC, Mul *	Sin:Mul 27:16		I–V	ER 17, RT 12, S 8, CTx 4, ER + RT 4, S + RT 1, WW 5	Rec 2, DOC 2	3.8 ^‡^	[40]
*n* = 8		Re/C, Mul ^†^	Sin:Mul 4:4		I–II	ER 2, S 4, CTx 1, S + CTx 1	CR	9 ^‡^	[42]
64	F	C	Sin	NA	Lo	EMR	CR	6	[43]
80	F	C + A	Mul	NA	Lo	E (HP+)	PR→PD	0.5	[86]
61	M	A	Sin	5	Lo	EMR	CR	NA	[107]
79	M	T	Sin	20	I	ESD + E (HP+)	CR	1	[73]
64	M	T	Sin	NA	IIE	S	CR	1	[71]
59	M	Sig	Sin	20	Lo	EMR	CR	3	[106]
54	M	Sig	Sin	20	IE	EMR	CR	0.8	[70]
50	F	Sig	Sin	18	Lo	S	NA	NA	[43]
83	F	AV25cm	Sin	NA	IE	RT→ER	CR	NA	[110]
57	F	Re	Sin	NA	Lo	ESD	NA	NA	[43]
58	F	Re	Sin	5	Lo	ESD	CR	0.8	[108]
54	F	Re	Sin	30	IE	EMR→ESD for residual tumor	CR	4	[109]
57	F	Re	Sin	>30	Lo	RT	CR	0.8	[43]
65	M	Re	Sin	10	I	RT	CR	5.4	[94]
83	F	Re	Sin	30	II2	R	CR	0.3	[39]
78	F	Re	Sin	30	NA	E	NC	3	[72]
53	F	Re	Sin	20	I	E (HP−)	CR	0.3	[87]
56	F	Re	Mul	10, 25	I	EMR + RT	CR	6.3	[94]
62	F	Re	Mul	6, 20	I	RT	CR	1.1	[94]

A: ascending colon; AV: anal verge; C: cecum; CR: complete remission; CTx: chemotherapy; DOC: died of other cause; E: eradication; EMR: endoscopic mucosal resection; ER: endoscopic resection; ESD: endoscopic submucosal dissection; F: female; FU: follow-up; He: hepatic flexure; HP: Helicobacter pylori; I: ileum; IC: ileocecum; Lo: localized; M: male; Mul: multiple; NA: not available; NC: no change; PD: progressive disease; PR: partial response; R: rituximab; Re: rectum; Rec: recurrence; RT: radiotherapy; S: surgery; Sig: sigmoid colon; Sin: single; T: transverse colon; Tx: treatment; WW: watch and wait. * Re 20, C 12, IC 15, Mul 4. ^†^ Re/C 7, Mul 1. ^‡^ Median follow-up time.

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
