# Peer review of "Mucosa-Associated Lymphoid Tissue (MALT) Lymphoma in the Gastrointestinal Tract in the Modern Era"

_cancers, 2022, doi:10.3390/cancers14020446_

Round 1
Reviewer 1 Report
The manuscript comprises a well written, comprehensive review on MALT lymphoma in the gastrointestinal tract. The analysis reflects in detail the pathogenesis, the molecular and pathological findings, different treatment modalities and their outcome in gastrointestinal MALT lymphoma.
The authors should add the following aspects with regard to radiotherapy and should add the following references:
-The German Study Group on Gastrointestinal Lymphoma (DSGL) aimed at decreasing radiation field size
-The DSGL reported the largest cohort on radiotherapy in gastric marginal zone lymphoma MALT, comprising 290 patients and analysed the potential radiogenic toxicity. And the DSGL reported the largest cohort in intestinal lymphoma, 134 patients, treated with radiotherapy.
Reinartz, G.; Pyra, R.P.; Lenz, G.; Liersch, R.; Stüben, G.; Micke, O.; Willborn, K.; Hess, C.F.; Probst, A.; Fietkau, R.; et al. Favorable radiation field decrease in gastric marginal zone lymphoma: Experience of the German Study Group on Gastrointestinal Lymphoma (DSGL). Strahlenther. Onkol. 2019, 195, 544–557, doi:10.1007/s00066-019-01446-5.
Reinartz, G.; Baehr, A.; Kittel, C.; Oertel, M.; Haverkamp, U.; Eich, H.T. Biophysical Analysis of Acute and Late Toxicity of Radiotherapy in Gastric Marginal Zone Lymphoma-Impact of Radiation Dose and Planning Target Volume. Cancers (Basel) 2021, 13, doi:10.3390/cancers13061390.
Ruebe C, Fischbach W, Bentz M, Daum S, Pott C, Tiemann M, Moeller P, Neubauer A, Wilhelm M, Lenz G, Berdel WE, Willich N, Eich HT. Renaissance of Radiotherapy in Intestinal Lymphoma? 10-Year Efficacy and Tolerance in Multimodal Treatment of 134 Patients: Follow-up of Two German Multicenter Consecutive Prospective Phase II Trials. Oncologist. 2020 May;25(5):e816-e832. doi: 10.1634/theoncologist.2019-0783. Epub 2020 Mar 27. PMID: 32219909; PMCID: PMC7216456. The authors also should describe and refer to the current multiinstitutional study from ILROG and GLA, short title 'GDL-ISRT 20 Gy', sudy center in Muenster, Germany, proving 20 Gy ISRT and additional biomarkers in gastric or duodenal lymphoma, see clinical trials.gov. NCT04097067Author Response
Thank you very much for the excellent comment. We added sentences in line 348-351 and line 354-359 in the revised version as follows:
line 348-351:
The German Study Group on Gastrointestinal Lymphoma (DSGL) reported that stage-adapted reduction from extended field RT (EFRT) to involved field RT (IFRT) led to a better outcome and reduction of adverse effects in gastric MALT lymphoma.
line 354-359:
Reinartz et al. showed that RT adapted to stage, histology, and resection in multimodal treatment of intestinal lymphoma achieved excellent local tumor control and survival rates despite partially decreasing field size [96]. In addition, current involved site RT (ISRT) offers the option of further reduction of normal tissue complication probability and a clinical trial of ISRT 20Gy for indolent localized gastrointestinal lymphoma is ongoing.

Reviewer 2 Report
This is a comprehensive review of intestinal MALT lymphoma.
In the introduction the authors should also include the association between lung MALT lymphoma and Achromobacter. They should also indicate that the association between some MALT lymphomas and the infective organism appears to show geographic variation.
The association between Helicobacter and gastric MALT lymphoma has decreased very markedly recently, particularly in the Western world with instances of Hp positive cases reported to be as low as 33% in a series from the UK. This does not come across well and the fact that the majority of cases are now Hp negative, at least in some areas should be more emphasised.
In the histology section is should be stated that lymphoepithelial lesions are not essential for the diagnosis of MALT lymphoma and that they can be seen in other low grade B cell lymphomas.
Also the distinction between MALT lymphoma and follicular lymphoma can be problematic, particularly if the biopsy is small and the MALT lymphoma shows is follicular collonisation associated with up-regulation of CD10 and/or bcl-6 in the intrafollicular compartment. in these instances FISH for the t(14;18) may be helpful.
The authors should also mention the potential role for very low dose radiotherapy (4Gy) which has been shown to be effective marginal zone lymphomas including those at extranodal sites.
Author Response
In the introduction the authors should also include the association between lung MALT lymphoma and Achromobacter. They should also indicate that the association between some MALT lymphomas and the infective organism appears to show geographic variation.
→Thank you very much for the comment. We added the association between lung MALT lymphoma and Achromobacter xylosoxidans in line 57 in the revised version. In addition, as reviewer pointed out, the association between bacterial infections and MALT lymphoma is variable in different geographical areas. We added the sentence in line 58 in the revised version as follows:
Other bacterial infections have been found to be implicated in the pathogenesis of MALT lymphoma arising in the skin (Borrelia burgdorferi), ocular adnexa (Chlamydophilia psittaci), lung (Achromobacter xylosoxidans), and the small intestine (Campylobacter jejuni). This association is variable in different geographical areas [3-5].
The association between Helicobacter and gastric MALT lymphoma has decreased very markedly recently, particularly in the Western world with instances of Hp positive cases reported to be as low as 33% in a series from the UK. This does not come across well and the fact that the majority of cases are now Hp negative, at least in some areas should be more emphasised.
→Thank you very much for the comment. We added the sentences in line 90-92, 287-288, and 469-471 to emphasize a significant decrease of the association between H. pylori and gastric MALT lymphoma as follow:
line 90-92:
The association between H. pylori and gastric MALT lymphoma has decreased recently, particularly in the Western world with instances of H. pylori-positive cases reported to be as low as 33% [17].
line 287-288:
Although a significant decrease in the rate of gastric MALT lymphoma associated with H. pylori infection have been reported, the effectiveness of eradication therapy can be seen even in H. pylori-negative cases, accounting for 0-57%, though the CR rate is lower than that of H. pylori-positive patients [8,28,62,82].
line 469-471:
Many patients with gastric MALT lymphoma respond to H. pylori eradication, although the association between H. pylori and gastric MALT lymphoma has decreased very markedly recently.
In the histology section is should be stated that lymphoepithelial lesions are not essential for the diagnosis of MALT lymphoma and that they can be seen in other low grade B cell lymphomas.
→Thank you very much for the comment. We added the sentences in line 182-183 in the revised version as follow:
The presence of LELs is suggestive of MALT lymphoma in the stomach, but is not essential for this diagnosis because they can be also seen in other low-grade B-cell lymphomas.
Also the distinction between MALT lymphoma and follicular lymphoma can be problematic, particularly if the biopsy is small and the MALT lymphoma shows is follicular collonisation associated with up-regulation of CD10 and/or bcl-6 in the intrafollicular compartment. in these instances FISH for the t(14;18) may be helpful.
→Thank you very much for the excellent comment. We added the sentences in line 196-201 in the revised version as follow:
The distinction between MALT lymphoma and follicular lymphoma can be problematic, particularly if the biopsy is small and the MALT lymphoma shows is follicular colonization associated with up-regulation of CD10 and/or BCL6 in the intrafollicular compartment. In these instances, fluorescent in situ hybridization (FISH) for the t(14;18)/BCL2 gene rearrangement may be helpful.
The authors should also mention the potential role for very low dose radiotherapy (4Gy) which has been shown to be effective marginal zone lymphomas including those at extranodal sites.
→Thank you very much for the excellent comment. We added the sentence in line 360-362 in the revised version as follow:
Recently, very-low-dose RT (4Gy) was also shown to be effective in MZLs including those at extranodal sites, which may alternatively reduce toxicities and duration of treatment [97].
